

# Comprehensive transcriptional analysis of pig facial skin development

Yujing Li[1,*], Rui Shi[1,*], Rong Yuan[2] and Yanzhi Jiang[1]

[1] Department of Zoology, College of Life Science, Sichuan Agricultural University, Ya'an, Sichuan, China
[2] Chengdu Livestock and Poultry Genetic Resources Protection Center, Chengdu, Sichuan, China
[*] These authors contributed equally to this work.

## ABSTRACT

**Background**. Skin development is a complex process that is influenced by many factors. Pig skin is used as an ideal material for xenografts because it is more anatomically and physiologically similar to human skin. It has been shown that the skin development of different pig breeds is different, and some Chinese pig breeds have the characteristics of skin thickness and facial skin folds, but the specific regulatory mechanism of this skin development is not yet clear.

**Methods**. In this study, the facial skin of Chenghua sows in the four developmental stages of postnatal Day 3 (D3) , Day 90 (D90) , Day 180 (D180), and Year 3 (Y3) were used as experimental materials, and RNA sequencing (RNA–seq) analysis was used to explore the changes in RNA expression in skin development at the four developmental stages, determine the differentially expressed messenger RNAs (mRNAs), long noncoding RNAs (lncRNAs), microRNAs (miRNAs), and circular RNAs (circRNAs), and perform functional analysis of related genes by Gene Ontology (GO) enrichment and Kyoto Encyclopedia of Genes and Genomes (KEGG) pathway analyses.

**Results**. A pairwise comparison of the four developmental stages identified several differentially expressed genes (DEGs) and found that the number of differentially expressed RNAs (DE RNAs) increased with increasing developmental time intervals. Elastin (ELN) is an important component of the skin. Its content affects the relaxation of the epidermis and dermal connection, and its expression is continuously downregulated during the four developmental stages. The functions of DEGs at different developmental stages were examined by performing GO and KEGG analyses, and the GO terms and enrichment pathways of mRNAs, lncRNAs, miRNAs, and circRNAs highly overlapped, among which the PPAR signaling pathway, a classical pathway for skin development, was enriched by DEGs of D3 *vs*. D180, D90 *vs*. D180 and D180 *vs*. Y3. In addition, we constructed lncRNA-miRNA-mRNA and circRNA-miRNA interaction networks and found genes that may be associated with skin development, but their interactions need further study.

**Conclusions**. We identified a number of genes associated with skin development, performed functional analyses on some important DEGs and constructed interaction networks that facilitate further studies of skin development.

Corresponding author
Yanzhi Jiang, jiangyz04@163.com

## INTRODUCTION

The skin is the largest organ of mammals and plays an important barrier role in protecting the internal environment of the body and resisting the invasion of pathogenic bacteria from the external environment (*Ai et al., 2014*). Because pig skin is anatomically and physiologically more similar to human skin than the skin of small mammals, such as rabbits, mice, rats and guinea pigs, it is used as an ideal material for modern medical xenografts (*Chen, Han & Zhang, 2002*; *Sullivan et al., 2001*). However, due to the differences in individuals and body parts, the characteristics of skin development after birth in different pig populations are different, such as skin thickness and wrinkles (*Chen & Wang, 1993*). Therefore, a clear understanding of the regulatory mechanism affecting pig skin development will enhance our understanding of pig skin to improve its applications.

RNA sequencing (RNA–seq) has been widely used to evaluate gene expression patterns in different species or in different stages of growth in the same species and has been used in combination with other disciplines, such as differential gene expression analysis and noncoding RNA (ncRNA) analysis (*Jia et al., 2020*; *Schliebner et al., 2014*; *Zhao et al., 2020*). NcRNAs constitute a class of RNAs that cannot be used as a translation template to synthesize proteins, including transfer RNAs (tRNAs), microRNAs (miRNAs), circular RNAs (circRNAs), and long noncoding RNAs (lncRNAs) (*Yang et al., 2016*). For a long time, ncRNAs have been considered to lack effective open reading frames (OFRs) and have no coding function, but in recent years, it has been gradually confirmed that some ncRNAs can encode functional peptides to participate in life activities, and it has been confirmed by sequencing and mass spectrometry that such ncRNAs and peptides generally have highly conserved and homologous properties (*Schmitz, Grote & Herrmann, 2016*; *Wade & Grainger, 2014*).

Skin development is a complex process that is influenced by many factors (*Zhu et al., 2020*). MicroRNA is the most studied endogenous small molecule RNA, and the length of mature miRNA is only approximately 18–24 nucleotides (nt), which can achieve posttranscriptional level regulation by inhibiting mRNA translation or promoting mRNA degradation (*He & Hannon, 2004*; *Rajewsky, 2006*). Meanwhile, miRNAs have been shown to influence cell proliferation, growth, and metabolism (*Sayed & Abdellatif, 2011*). In addition, miRNAs play a vital regulatory role in the skin; for example, small extracellular vesicles from dermal fibroblasts can promote fibroblast activity by carrying miRNA-218 and then affect the development of skin (*Zou et al., 2022*). LncRNAs are generally longer than 200 nt, which is close to the length of some mRNAs (*Mercer, Dinger & Mattick, 2009*). Several studies have confirmed that lncRNAs are involved in different biological processes in skin development (*Fan, Huang & Chen, 2021*; *Ren et al., 2016*; *Zhu et al., 2020*). CircRNA is a special class of single-stranded RNAs with a circular covalent closed-loop structure with diverse functions. CircRNAs can not only enrich miRNA to achieve posttranscriptional regulation but also form complexes with RNA-binding proteins, regulate the expression of parental genes, and even participate in protein translation as a template (*Huang et al., 2016*; *Liang & Wilusz, 2014*). Studies have shown that circ004463 is associated with fibroblast proliferation and collagen I synthesis during skin development (*Zou et al., 2023*). RNA-seq

has been used in skin studies of many species in recent years, revealing the underlying mechanisms of skin development. However, research on pig skin is still relatively scarce and has broad prospects.

In the present study, we identified differentially expressed genes (DEGs) at different developmental stages by the expression profiles of mRNAs, miRNAs, lncRNAs, and circRNAs in skin tissue at D3, D90, D180, and Y3. Some genes and pathways that may be related to skin growth and development were obtained, which provided some basis for further study of skin development and revealed its related regulatory mechanisms.

## MATERIALS & METHODS

### Animals and tissue collection

In this study, 12 Chenghua sows from four different development periods, including 3-days-old, 90-days-old, 180-days-old, and 3-years-old, were selected as experimental animals, and three pigs from the same litter were considered as the biological replicates per development period. All pigs were divided into four experimental groups according to the development periods, which included the D3 group, D90 group, D180 group and the Y3 group. The piglets were weaned at the age of $28 \pm 1$ day. A starter diet containing of 18.0% crude protein, 7.0% crude ash and 1.32% lysine was administered to piglets from day 30 to day 45 after weaning. From day 46 to day 179, the pig's diet consisted of 16.0% crude protein, 9.0% crude ash and 1.0% lysine. From the 180st day, pigs received a diet containing of 14.0% crude protein, 9.0% crude ash and 0.6% lysine. Pigs are allowed free access to food and water under the same conditions. Food and water were withheld from the pigs for 24 h before slaughter. After being transported to the site of slaughter, they were allowed to rest for 2 h and then were humanely slaughtering. To reduce pain, a 10-second sudden shock at 50 V and 90 Hz was used.

The 12 Chenghua sows in the experiment were sourced from the Chengdu Livestock and Poultry Genetic Resources Protection Center in Sichuan Province, China. All animal experimental procedures were approved by the Institutional Animal Care and Use Committee of Sichuan Agricultural University (permit number: 20220279).

The facial skin tissue of each pig and the fresh samples were flash-frozen in liquid nitrogen and then stored at −80 °C until RNA was extracted. TRIzol Reagent (Invitrogen, Waltham, MA, USA) was used to extract total RNA, which was subsequently treated with DNase and purified using an RNeasy Mini Kit (Qiagen, Valencia, CA, USA). The quality, concentration, and integrity of RNA were checked using a nanodrop photometer and an Agilent 2100 bioanalyzer.

### RNA library construction and sequencing

The lncRNA library includes mRNA and lncRNA. According to the manufacturer's information, the extracted total RNA was first removed using the MGIEasy rRNA Removal Kit. Then, it was submitted to RNA fragmentation and cDNA synthesis (second-strand cDNA synthesis with dUTP instead of dTTP), followed by end repair, addition of an A residue to the 3′ end and adapter ligation, PCR, circularization and generation of DNA

Nano Ball (DNB), and finally sequencing on the DNBSEQ-G400 platform and 150 bp paired-end reads were obtained.

Once the Small RNA library was generated, RNAs of 18–30 nt in length were purified and separated using Polyacrylamide Gel Electrophoresis (PAGE) for 3′ and 5′ linkages. Reverse transcription extension with RT primers with Unique Molecular Identifier (UMI)was use to synthesize cDNA. PCR amplification of cDNA was performed with both 3′ and 5′ linkers linked to highly sensitive polymerase to amplify yield. PCR products in the 110–130 bp range were separated using PAGE. Library quantification, pooling cyclization, and quality inspection of the constructed library were performed. Libraries that passed the quality test were sequenced on the DNBSEQ-G400 and 50 bp single-end reads were obtained.

CircRNAs were constructed using DNase I to digest the DNA fragments present in the total RNA sample; then, they were purified and recycled, and ribosomal RNAs were removed from the total RNA sample using the Ribo-off method. The RNase R reaction system was prepared, linear RNA components were digested, reaction products were purified and recovered, RNA was fragmented at a certain temperature and ion environment, one-stranded cDNA was synthesized using fragmented RNA as a template, and two-stranded cDNA was synthesized with dUTP instead of dTTP. The ends of the double-stranded cDNA were repaired, an ''A'' residue was added to the 3′ end, the linker was ligated, and the two-stranded cDNA containing ''U'' residues was digested with UDGase and then submitted to PCR amplification. The constructed library was quality tested, the library products that passed the quality test were cycled, and the circular DNA molecules were copied through rolling rings to form DNA nano balls (DNBs), and finally sequenced by using the DNBSEQ-G400 platform, obtain 150bp paired-end reads.

## Quality control and read alignment

After removing the reads containing the adaptor (adaptor pollution), low-quality reads and reads whose N content was greater than 5% from the raw sequencing data, the resulting clean reads were compared to the reference genome and transcriptome (GCF_000003025.6_Sscrofa11.1).

## Expression quantification

We used Bowtie2 (*Langmead & Salzberg, 2012*) to map the lncRNA library clean reads to the reference sequence and then used RSEM (*Li & Dewey, 2011*) to calculate the expression levels of genes and transcripts. Similarly, we used Bowtie2 to align clean reads to the reference set and other small RNA databases and used UMI for relevant gene expression calculations. Meanwhile, we calculated the expression of circRNA based on the number of back-spliced reads compared to both ends of the circRNA and used two software programs, CIRI (https://sourceforge.net/projects/ciri) and find_circ (https://github.com/marvin-jens/find_circ), prediction. The final back-spliced read number is the average of the two results.

## Differential expression analysis

The four developmental stages Chenghua sows had three biological replicates, respectively. The statistical power of this experimental design, calculated in RNASeqPower is 0.84

(https://doi.org/doi:10.18129/B9.bioc.RNASeqPower). A differential expression analysis of RNA was performed using the DEGseq R package. DEGseq (*Wang et al., 2010*) based on MA-plot (*Yang et al., 2002*) was used to calculate the differential expression. The *p*-value was adjusted to the *q*-value by *Benjamini & Hochberg (1995)* and *Storey & Tibshirani (2003)*. A *Q*-value <0.005 and |log2 (fold change)|>1 were set as the thresholds for significantly differential expression.

According to the results of differential gene detection, the R package heatmap was used to perform hierarchical clustering analysis on the union set differential genes (https://cran.r-project.org/web/packages/pheatmap/). When multiple groups of DE miRNAs were clustered at the same time, we performed separate cluster analyses on the intergroup intersection and union of differentially expressed miRNAs.

Based on the GO and KEGG annotation results and the official classification, we performed functional classification and biological pathway analysis of genes derived from different circRNA sources, while enrichment analysis was performed using the hyperfunction in R software. The *p*-value was then adjusted to the *q*-value, and typically pathways with *q*-value <= 0.01 are considered to be significantly enriched.

## Target gene prediction of lncRNAs and miRNAs

For differentially expressed lncRNAs, we predicted their target genes by the following steps: calculating the spearman and pearson correlation coefficients between lncRNAs and mRNAs and requiring spearman coefficient >=0.6 and pearson coefficient >=0.6. Then, it was considered the position relationship between lncRNAs and mRNAs. When the lncRNA was located within 20Kb upstream and downstream of the mRNA, it was considered to be cis-regulated. Beyond this range, it would analyze the binding energy of lncRNA and mRNA by RNAplex (v0.2) (*Tafer & Hofacker, 2008*), and when the binding energy is <-30 and it would be judged as tran-acting mode. RNAhybrid (*Kruger & Rehmsmeier, 2006*), miRanda (*John et al., 2004*) and TargetScan (*Agarwal et al., 2015*) were used to predict target genes of differentially expressed miRNAs.

## RNA interaction network construction

The target genes of differentially expressed lncRNAs and circRNAs were predicted by miRanda and RNAhybrid software, and the target genes predicted by the two software programs were selected. Finally, Cytoscape 3.9.1 (http://cytoscape.org/) was used to map the corresponding RNA interaction network according to the predicted lncRNA−miRNA, miRNA−mRNA, and circRNA−miRNA.

## Hematoxylin and Eosin (H&E) Staining

According to the manufacturer's instructions, fixed tissues were embedded in paraffin (Servicebio, Wuhan, China) and cut into 3–4 μm thick sections using a microtome (Servicebio, Wuhan, China). Images were captured with a microscope (Nikon, Japan). All staining assays were performed in triplicate.

## Validation Real-Time qPCR (qRT − PCR)

After RNA isolation, cDNA is synthesized by a reverse transcription kit (Takara, Chengdu, China). qRT −PCR was performed with SYBR-Green I nucleic acid dye in the CFX 96

real-time system (BIO-RAD, USA). Primer sequences for 2 mRNAs, 2 lncRNAs, 1 miRNA, and 1 circRNA were designed and synthesized by NCBI and Primer 5 (Table S1). GAPDH was used to normalize the expression levels of mRNAs, lncRNAs and circRNAs, and U6 was used to normalize the expression levels of miRNAs. At least three samples were analyzed for each developmental stage (D3, D90, D180, and Y3), and each sample was analyzed in three independent reactions. The results were statistically analyzed using $2-\Delta\Delta$ CT relative quantification.

## RESULTS

### Descriptive statistics and correlation analysis of skin development in pigs after birth

To understand the development of skin in pigs after birth, we performed RNA-seq on the facial skin of postnatal Chenghua pigs (D3, D90, D180, and Y3). H&E staining of the facial skin tissue of Chenghua pigs in four developmental stages showed that the skin thickness in the slices increased with increasing age (Fig. 1A). In this study, we constructed lncRNA and small RNA libraries. A total of 44,611 genes were detected in the lncRNA library (Table S2), and the average alignment rate of the sample com parison genome was 90.35% (Table S3). A total of 1,769 small RNAs were detected in the small RNA library (Table S4), and the average comparison rate of the sample comparison genome was 90.25% (Table S5). The expression of circRNA is shown in Table S6. The proportion of clean reads Q30 (or Q20) of the filtered RNA was greater than 92% (Tables S7–S9), and the base mass distribution of clean reads showed a low proportion of bases of low quality (Quality<20) (Fig. S1–S3), which indicates good sequencing quality.

Principal component analysis (PCA) was used for the identified mRNA and lncRNA transcripts, and the results showed that the D90 group relative aggregation, D3, and Y3 had a very low correlation, and the similarity of adjacent developmental stages was higher than that of nonadjacent developmental stages (Fig. 1B). According to the expression boxplot, the distribution of miRNA expression levels in the four periods was concentrated, especially at D180 (Fig. 1C). At the same time, the PCA results of miRNA also showed that the correlation between D3 and Y3 was lower than that between D3 and D90 and between D3 and D180, and the similarity of adjacent developmental stages was higher than that of nonadjacent developmental stages. The D90 group was clustered together, which was similar to the PCA results of mRNA and lncRNA (Fig. 1D).

After the quality control of transcriptome sequencing, the RNAs expression profiles at different time points were determined, and mRNA was confirmed to be expressed by 17,998, 17,957, 17,882, and 17,819 genes; miRNA was confirmed to be expressed by 1,447, 1,342, 1,297, and 1,294 genes; lncRNA was confirmed to be expressed by 16,392, 16,662, 16,454, and 16,108 genes; and circRNA was confirmed to be expressed by 6,260, 6,373, 6,243, and 6,043 genes at D3, D90, D180, and Y3, respectively (Fig. 1E).

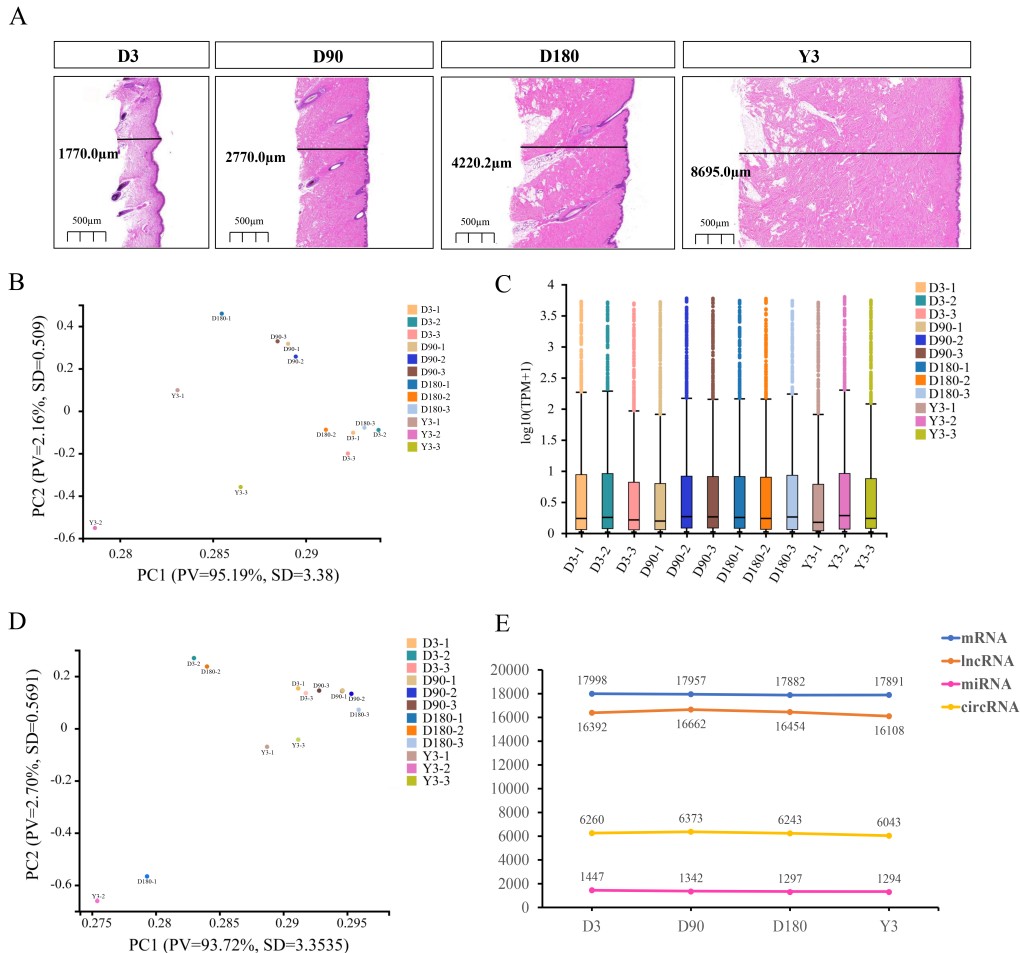

**Figure 1** **Descriptive statistical analysis of pig skin development during D3, D90, D180, and Y3 stages.**
(A) Skin thickness at four developmental stages. Principal Component Analysis (PCA) plot of identified
mRNAs, lncRNAs, (B) and miRNAs (D). PV means "Proportion of variance", SD means standard devi-
ation. (C) Boxplot of mRNAs and lncRNAs expression quantity. (E) The number of mRNAs, lncRNAs,
miRNAs, and circRNAs expressed during the four developmental stages.

## Identification of differentially expressed mRNAs and noncoding RNAs (ncRNAs)

The molecular mechanism and pathway of pig skin growth and development are related to
the expression abundance of specific RNAs. Statistics on the basal expression of mRNAs,
lncRNAs, miRNAs and circRNAs at the four developmental stages showed that although
there were tens of thousands of RNAs expressed in each stage, the proportion of high-
expression genes that truly played a role in skin development was not high (Table 1).
Interestingly, the genes with the highest expression for all four types of RNA at the four-
time points were basically the same.

In this study, in skin tissue, a total of 242, 428, 625, 54, 323, and 138 upregulated and
220, 446, 453, 54, 114, and 41 downregulated mRNA differentially expressed genes (DEGs)
(Fig. 2A); 38, 65, 39, 23, 18, and 20 upregulated and 41, 76, 78, 37, 55, and 41 downregulated

Li et al. (2023), *PeerJ*, DOI 10.7717/peerj.15955

**Table 1 Expression statistics of mRNAs, lncRNAs, miRNAs, and circRNAs.**

| expression | mRNA | | | | lncRNA | | | | miRNA | | | | circRNA | | | |
|---|---|---|---|---|---|---|---|---|---|---|---|---|---|---|---|---|
| | D3 | D90 | D180 | Y3 | D3 | D90 | D180 | Y3 | D3 | D90 | D180 | Y3 | D3 | D90 | D180 | Y3 |
| (0, 1) | 3821 | 3624 | 3703 | 3940 | 14683 | 14496 | 14684 | 14565 | 930 | 850 | 818 | 842 | 0 | 0 | 0 | 0 |
| [1, 10) | 5182 | 4980 | 5274 | 5430 | 1556 | 1941 | 1622 | 1425 | 259 | 270 | 256 | 239 | 0 | 0 | 0 | 0 |
| [10, 100) | 7998 | 8425 | 8026 | 7527 | 131 | 166 | 139 | 111 | 140 | 101 | 107 | 96 | 4314 | 4296 | 4307 | 4153 |
| [100, 1000) | 883 | 852 | 786 | 838 | 18 | 17 | 12 | 10 | 70 | 70 | 71 | 81 | 1820 | 1951 | 1802 | 1754 |
| [1000, $+\infty$) | 114 | 76 | 93 | 85 | 3 | 2 | 2 | 2 | 48 | 51 | 45 | 36 | 126 | 126 | 134 | 136 |

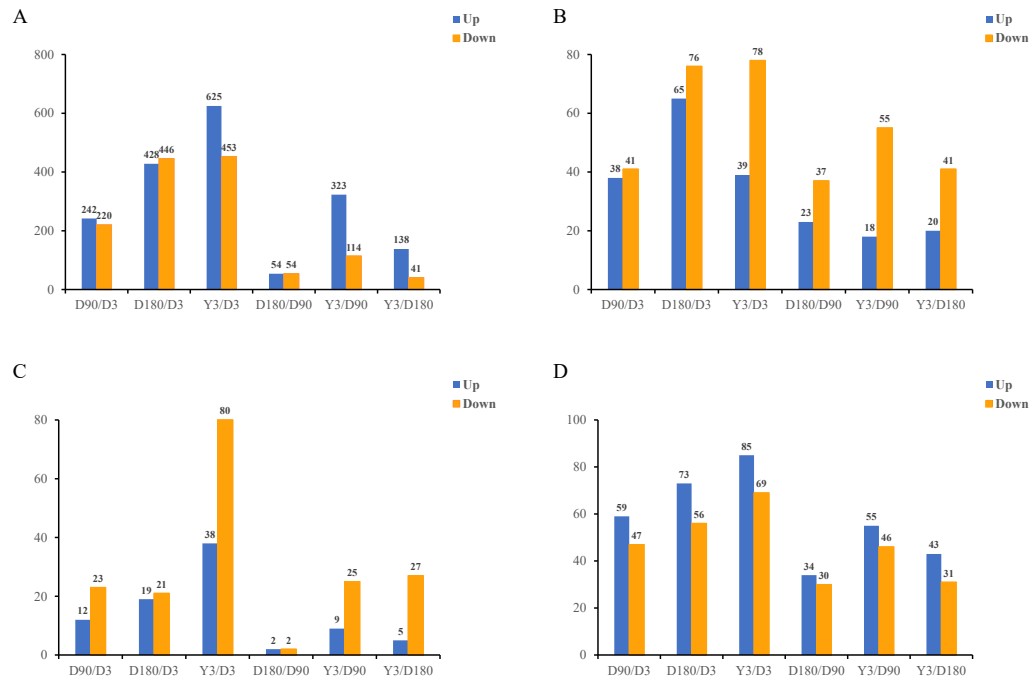

**Figure 2** Identification of differentially expressed mRNAs (A), lncRNAs (B), miRNAs (C), and circR-NAs (D) at different developmental stages.

lncRNA DEGs (Fig. 2B); 12, 19, 38, two, nine and five upregulated and 23, 21, 80, 2, 25, and 27 downregulated miRNA DEGs (Fig. 2C); and 59, 73, 85, 34, 55, and 43 upregulated and 47, 56, 69, 30, 46, and 31 downregulated circRNA DEGs (Fig. 2D) were detected in D3 *vs.* D90, D3 *vs.* D180, D3 *vs.* Y3, D90 *vs.* D180, D90 *vs.* Y3 and D180 *vs.* Y3 ($|\log2FC|>1$, Q value $<0.05$). Interestingly, as the time interval between development increases, so does the number of differentially expressed RNAs. For example, D3 *vs.* Y3 had significantly more differentially expressed RNAs than D3 *vs.* D90.

## The function of the differentially expressed mRNAs

The Upset diagram clearly shows the expression of the differentially expressed mRNAs (DE mRNAs) between the different comparison groups of the four developmental stages. Two DE mRNAs (immunoglobulin superfamily member 10, IGSF10 and Elastin, ELN) were expressed in all comparison groups (Fig. 3A), especially ELN, which was relatively high in each stage, and the expression of the two genes was continuously downregulated (Fig. 3B). The results show that these coexpressed DE mRNAs may have important roles in skin growth and development, especially due to their expression levels and the fact that they are co-expressed.

Heat shock protein 70.2 (HSP70.2) and heat shock protein family B (small) member 1 (HSPB1) were identified as DEGs with upregulated mRNA expression in D3 *vs.* D180 and D90 *vs.* Y3, and keratin 31 (KRT31), keratin 34 (KRT34), keratin 33A (KRT33A), collagen type XIV alpha 1 chain (COL14A1), collagen type XXI alpha 1 chain (COL21A1), collagen
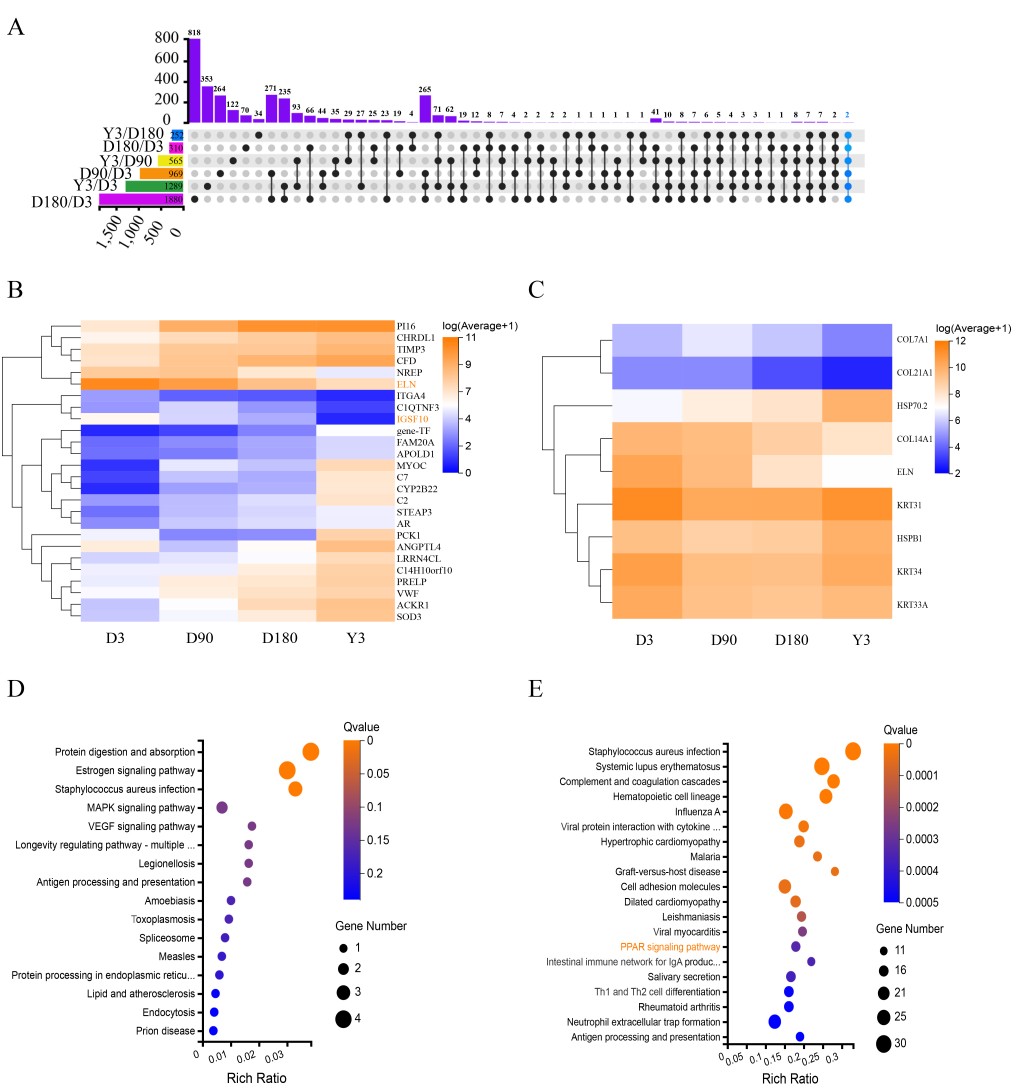

**Figure 3 Functional analysis of differentially expressed mRNAs.** (A) Upset diagram of DE mRNAs. (B) Heatmap of DEGs present in multiple comparison groups. (C) Heatmap of the 9 DEGs. Pseudo-colors show expression levels from orange (high) to blue (low). Enrichment analysis of KEGG pathway for 9 DEGs (D) and D3 *vs.* D180 DEGs (E). DE mRNAs, differentially expressed mRNAs; DEGs, differentially expressed genes; KEGG, Kyoto Encyclopedia of Genes and Genomes.

type VII alpha 1 chain (COL7A1) and ELN were identified as DEGs with downregulated mRNA expression in D3 *vs.* D180, D3 *vs.* Y3, D90 *vs.* D180 and D90 *vs.* Y3. As shown in the heatmap, the expression levels of these DE mRNAs were more significant (Fig. 3C). According to the KEGG pathway analysis, we found that these genes were mainly enriched in the following pathways: protein digestion and absorption, estrogen signaling pathway, and staphylococcus aureus infection (Fig. 3D). In addition, KEGG analysis of DEGs for D3 *vs.* D180 found that they were significantly enriched in the skin growth and development-related pathway such as peroxisome proliferators-activated receptors (PPAR)

(Fig. 3E), and DEGs from D90 *vs.* D180 and D180 *vs.* Y3 were also enriched in this pathway (Fig. S4).

## The function of target genes of differentially expressed noncoding RNAs (ncRNAs)

Gene Ontology (GO) analysis was used to analyze the main functions of the differentially expressed miRNAs (DE miRNAs). According to the GO database, we found that the terms enriched across groups of miRNAs had a high degree of overlap, and they were mainly enriched in the following biological processes: cellular process, biological regulation, and regulation of biological process and metabolic process; cellular components: cell, cell part, organelle and membrane; and molecular functions: binding, catalytic activity and molecular function regulator (Fig. 4A, Fig. S5). In addition, there was a high degree of overlap in KEGG pathways between the comparison groups, including cell growth and death, cellular community-eukaryotes, folding, sorting and degradation, development and regeneration, aging and other pathways (Fig. S6). This suggests that these terms may play an important role in the development of skin after birth.

To explore the function of lncRNA-associated mRNA, we selected the two genes (URS0000EF6FAF, URS0001974785) with the highest expression and 1 gene (URS0000EF1D34) with continuous upregulation from all upregulated lncRNAs and selected the top six genes with the highest expression from all downregulated lncRNAs, of which four genes (URS0001952842, URS0000EF04FE, URS0001977D95, and MEG3) were continuously downregulated, and one gene (URS0001961E18) showed no expression at the Y3 stage (Fig. 4B). We predicted the target genes of these nine lncRNAs and obtained 4 associated mRNAs. GO analysis found that the enriched terms were similar to those of miRNA (Fig. 4C).

The function of circRNA is related to the function of host linear transcripts, and we used GO and KEGG analyses to determine the host genes that the circRNAs differentially regulated. Through GO analysis, we found that each comparison group was mainly enriched in biological processes, such as cellular process, biological regulation and regulation of biological process, and developmental process, especially D3 *vs.* Y3, a comparison of highly separated developmental stages, for which 12 genes were enriched in developmental process (Fig. 4D, Fig. S7). In addition, KEGG analysis found that the differentially expressed circRNAs (DE circRNAs) had common differential signaling pathways, including cellular community-eukaryotes, cell growth and death, transport and catabolism, aging, and development (Fig. S8).

## Construction of RNA interaction networks

RNA transcripts communicate through the ceRNA language, and lncRNAs act as sponges for miRNAs to regulate gene expression (*Chen et al., 2019*; *Tay, Rinn & Pandolfi, 2014*). In this study, we constructed the lncRNA–miRNA–mRNA coexpression network through Cytoscape, consisting of nine lncRNA nodes, 32 miRNA nodes and 107 mRNA nodes (Fig. 5A). LncRNAs, such as URS0001977D95 and URS0001961E18, target 28 miRNAs respectively, and miRNAs, such as novel-ssc-miR1-5p and novel-ssc-miR107-5p, are

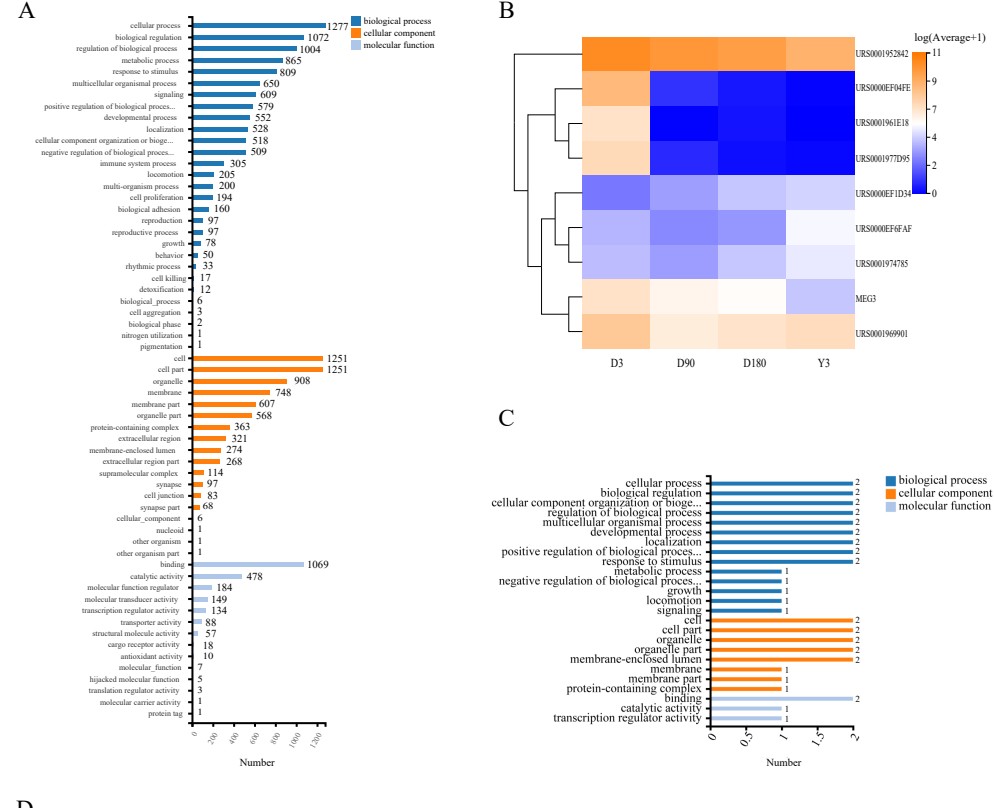

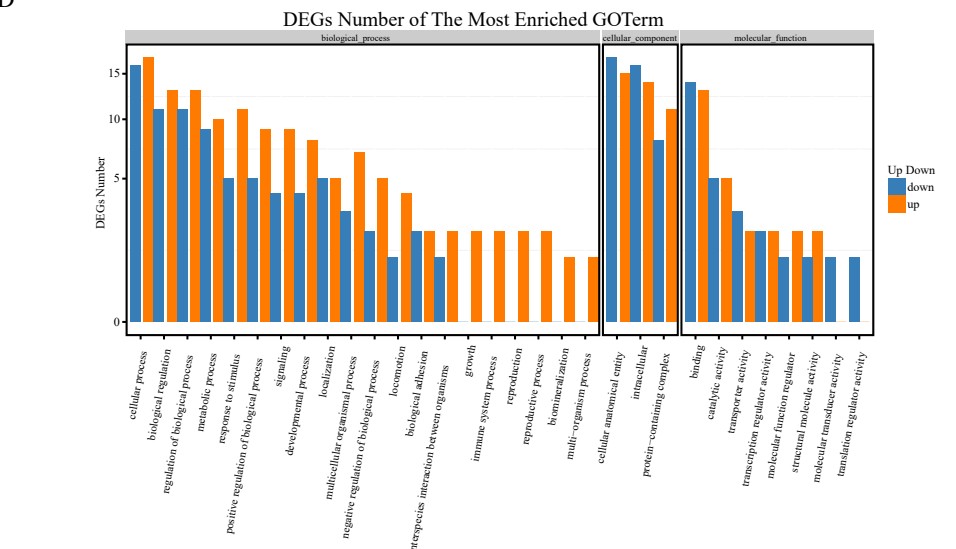

**Figure 4  Function of target genes of differentially expressed non-coding RNAs.** (A) The enriched GO terms of DE miRNAs. (B) Heatmap of the 9 DE lncRNAs. (C) The enriched GO terms of the 9 DE lncRNAs. (D) The enriched GO terms of DE miRNAs in D3 *vs.* Y3. GO, gene ontology; DE miRNAs, differentially expressed miRNAs; DE lncRNAs, differentially expressed lncRNAs; DE circRNAs, differentially expressed circRNAs.

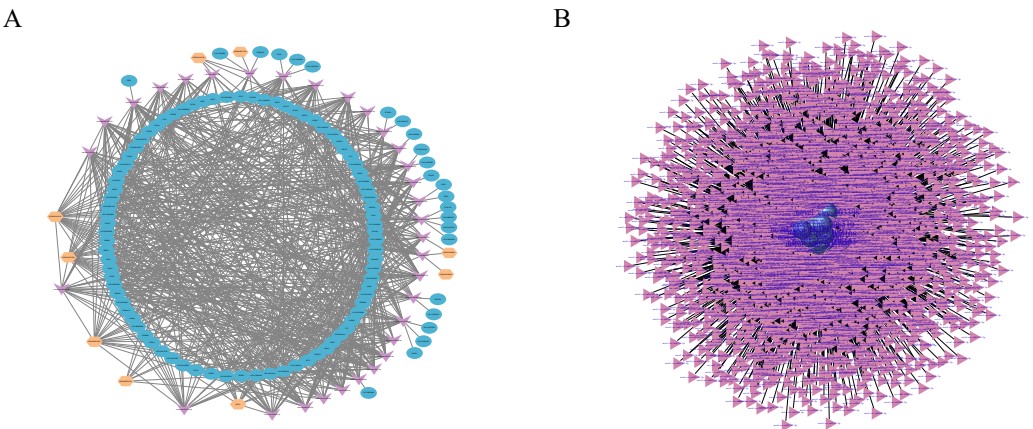

**Figure 5** **RNAs interaction networks.** (A) lncRNA-miRNA-mRNA interaction network. Orange, pink, and blue are representative of lncRNAs, miRNAs, and mRNAs, respectively. (B) circRNA–miRNA interaction network. Purple, and pink are representative of circRNAs, and miRNAs, respectively.

targeted by five lncRNAs. In addition, mRNAs, such as cyclin dependent kinase inhibitor 2B (CDKN2B) is associated with cell growth and death, and FERM domain containing 4B (FRMD4B) is associated with corpus callosum agenesis with facial anomalies and cerebellar ataxia.

Studies have shown that circRNAs can act as competitive endogenous RNAs (ceRNAs) to regulate miRNA function (*Schorr & Mangone, 2021*; *Zhang et al., 2021*), suggesting that circRNAs and their target miRNAs may be coexpressed in the development of skin tissues. Therefore, we used miRanda and RNAhybrid software to predict the target miRNAs of the circRNA. We identified a total of 13,584 circRNAs and 3,228 miRNAs with targeted binding relationships and constructed an interaction network diagram for the top 20 circRNAs with miRNAs with the greastet correlations (Fig. 5B). However, all these findings require further study.

### Validation of RNA-seq data

We randomly selected two mRNAs, two lncRNAs, one miRNA, and one circRNA to validate the whole transcriptome sequencing data by real-time quantitative PCR (qRT–PCR). The q–PCR results were consistent with the RNA-seq data (Fig. 6, Table S10). The expression of mRNA, such as CTSK and ELN, showed significant differences over time in the four groups. In addition, lncRNA (MEG3 and URS0001952842), miRNA (ssc-miR-30b-5p), and circRNA (novel-circ-008475) also showed the same differences. Therefore, the q −PCR results verified the accuracy of the RNA-seq data.

## DISCUSSION

Currently, some species, such as rats, sheep and pigs, are used in skin research, mainly in the fields of skin diseases, scarring and wound healing (*Hassanshahi et al., 2019*). Pigs are a common class of large mammals whose genetic makeup closely resembles that of humans, and they have individual and numerical advantages over other specie (*Plakhotnyi et al.,*

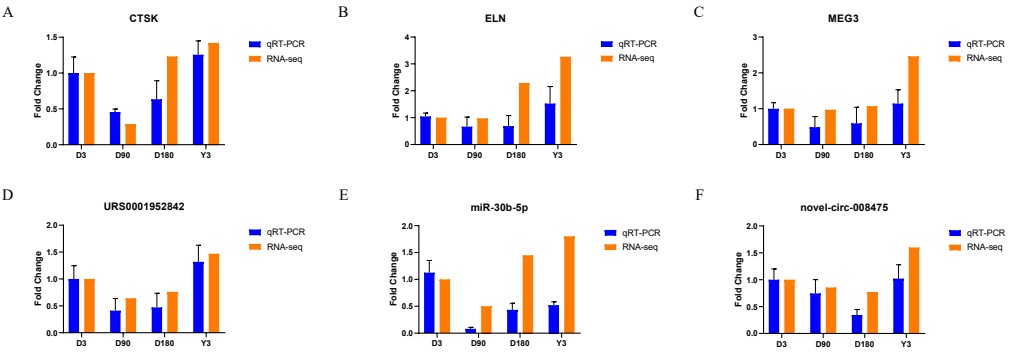

**Figure 6** Validation of mRNA (A–B), lncRNA (C–D), miRNA (E), and circRNA (F) data by qRT-PCR.

*2021*). Some pig breeds from all over the world, such as large white pigs, long white pigs, erhua face pigs and Bama Xiang pigs, have been studied. Among them, most studies focus on large white pigs and there are few studies on local pigs in China (*Roth et al., 2022*; *Zhao et al., 2021*). According to previous research, the Chenghua pig, a famous local pig breed in Southwest China, shows a specific skin thickness characteristic, and the thickest skin on its back can reach 8.0 mm and much higher than that of the foreign large white pig (*Li et al., 2022*; *Zou et al., 2022*).

Some studies have revealed that skin thickness shows significant differences among different age and part. For example, the skin thickness of adults will be between 0.5 mm and 4.0 mm depending on age and part (*Foster et al., 2000*), while the total thickness of children's skin is 0.92−2.2 mm. Meanwhile, the skin on the face is the thinnest part but the skin on the back and buttocks is the thickest parts (*Zhang, 2006*). Moreover, skin thickness is closely related to the appearance of human skin, such as sagging and wrinkling (*Qin et al., 2018*), and facial wrinkles appear in head, typically increasing along with aging (*Huang et al., 2019*). Here, hematoxylin and eosin (H&E) staining showed that the thicknesses of facial skin sections at D3, D90, D180, and Y3 were 1,770.0 μm, 2,770.0 μm, 4,220.2 μm and 8,695.0 μm, respectively. Therefore, with increasing age, skin thickness also increases. This is basically consistent with previous research results. At the same time, this phenotypic difference has become the fundamental basis for RNA-seq (*Arindrarto et al., 2021*; *van Dijk et al., 2018*).

In this study, RNA-seq was used to explore transcriptome sequencing analysis, including mRNA, lncRNA, miRNA, and circRNA. In the corresponding RNA library, we obtained clean reads by filtering raw reads and then compared these high-quality reads to the reference genome to obtain the total basal expression profile of the four types of RNA. The number of mRNAs, miRNAs, lncRNAs, and circRNAs compared to the reference genome were approximately 18,000, 1,300, 16,000, and 6,000, respectively. However, the numbers of genes with basal expression higher than 1,000 (TPM) accounted for only approximately 0.51%, 3.46%, 0.0125%, and 2.175% of mRNAs, miRNAs, lncRNAs, and circRNAs, respectively, which may indicate that miRNA and circRNA play an important

role in skin development due to their competitive regulation, although few of them have high expression (*Zhu et al., 2018*).

It has been reported that the collagen content of Chenghua pig skin is extremely rich, which makes it a suitable, although rare, source of skin research materials. The top ten genes with the highest expression of mRNA in the four stages of D3, D90, D180, and Y3 corresponded to the human genes collagen type III alpha 1 chain (COL3A1), testis expressed 50 (TEX50), collagen type I alpha 2 chain (COL1A2), testis expressed 14, intercellular bridge forming factor (TEX14), collagen type I alpha 1 chain (COL1A1), basic salivary proline-rich protein 4-like (LOC110258214), progesterone receptor-like (LOC110258215), basic salivary proline-rich protein 2-like (LOC110258600), eukaryotic translation elongation factor 1 alpha 1 (EEF1A1), and secreted protein acidic and cysteine rich (SPARC). Among them, COL3A1 is the major collagen comprising skin connective tissue (*Wang et al., 2007*). COL1A2 and COL1A1 are the most abundant collagens in many human tissues, such as bone, skin and tendons, and are related to skin growth and development, which is confirmed by the specific skin thickness trait of Chenghua pigs (*Lee et al., 2022*). Studies have shown that 9-methoxycanthine-6-one affects the expression of EEF1A1, and 9-methoxycanthine-6-one is related to the activity of anticancer substances *in vitro* for skin cancer (*Yunos et al., 2022*). SPARC is associated with accumulation of skin basement membrane and production of type IV and VII collagen (*Nakamura et al., 2022*). There are few literature reports about TEX50 and TEX14 related to skin, and the reason and potential role of their high expression in facial skin tissue need to be further studied. Similarly, LOC110258214, LOC110258215, and LOC110258600 are expressed in various organs, and their high expression levels in various stages of skin indicate that they play a certain role in the process of skin development, but the specific role needs further study.

By comparing the four time points of D3, D90, D180, and Y3, it was found that the number of DEGs of mRNA was the largest, and the number DEGs of the other three type of RNA was less than 100. Among the DEGs of mRNA, ELN is a gene that is continuously downregulated and has a high expression amount, and elastin is also an important part of the skin, and its reduced contentresults in loosening of the connection between the epidermis and dermis, which may be an important cause of skin aging and wrinkling during development and growth (*De Miranda, Weimer & Rossi, 2021*). At the same time, in the mRNA gene expression profiles of the comparison groups D3 *vs.* D180 and D90 *vs.* Y3, we found that the heat shock proteins HSP70.2 and HSPB1 were upregulated, they are important chaperones, are involved in cytoskeletal stability, cell migration, regulation of cell growth and differentiation, and are related to cell anti-apoptosis (*Kanagasabai et al., 2010*; *Lee et al., 2006*; *Ling et al., 2018*). The two major stress-inducible genes Hsp70-1 and Hsp70-2 were found to be upregulated in the allogeneic rat skin explant assays. And the MHC-encoded Hsp70-1 and Hsp70-2 genes might serve as new markers of GVHR, helping to further increase the predictive value of the skin explant assay (*Novota et al., 2008*). In addition, KRT31, KRT34, KRT33A, COL14A1, COL21A1, COL7A1, and ELN were found to be downregulated in the mRNA gene expression profiles of the comparison groups D3 *vs.* D180, D3 *vs.* Y3, D90 *vs.* D180 and D90 *vs.* Y3. Among them, keratin has the inherent ability to promote cell adhesion, proliferation and tissue regeneration, and its

biomaterials can provide biocompatible matrices for the regeneration of defective tissues (*Shavandi et al., 2017*). According to previous research, keratinocytes are located on the surface of the skin and their dysregulated innate immune response may lead to uncontrolled inflammation and psoriasis pathogenesis, implicated in skin healing. The down-regulation of KRT31, KRT34, and KRT33A may affect the regeneration of damaged skin (*Zhang, Yin & Zhang, 2019*). Collagen, an important component in the skin, plays a structural role in contributing to the mechanical properties, tissue structure, and tissue shape, and variants in COL7A1 may cause neonatal bullous dermolysis and allergic bullous epidermolysis (*Chao et al., 2022*; *Ricard-Blum, 2011*). COL14A1, COL21A1, and COL7A1 were down-regulated, which may be the reason why the skin gradually loses elasticity during skin development with age.

In KEGG analysis of mRNA, we found that for D3 *vs.* D180, genes were significantly enriched in the PPAR signaling pathway, which is associated with skin growth and development, is thought to be involved in skin barrier formation and fibroblast differentiation (*Ghosh, 2021*; *Sobolev et al., 2022*), and is also present in the enrichment results of D90 *vs.* D180 and D180 *vs.* Y3, suggesting that D180 may be a critical developmental inflection point.

The most studied ncRNA that regulates mRNA translation is miRNA. Among the most highly expressed mRNAs, ssc-let-7f-5p is expressed only at the D3 stage, and is involved in immune and embryonic development processes (*Hua et al., 2021*), indicating that may be related to the growth and development of the embryonic skin. In addition, ssc-miR-22-3p is associated with inflammation and is expressed in both D90 and D180 stages (*Swain et al., 2021*), which may be related to skin inflammation and skin diseases. This is also consistent with the growth and development of mammals after birth, but further research is needed to verify. There are relatively few studies on lncRNAs in skin development. URS0000EF6A54 (expression from D3 to Y3: 26,574, 27,576, 30,860, and 50,769) and URS0001979623 (expression from D3 to Y3: 52,373, 60,455, 73,781, and 63,325) were highly expressed at the four time points, and URS0000EF6A54 is a continuously upregulated gene, that may be related to growth and development, but its function is unknown. Further research is needed. Similar to lncRNAs, most circRNAs do not yet have specific functions. As the only nonlinear covalent ncRNA, the functional realization of circRNA comes more from binding multiple miRNAs at the same time, as in Table 1. No circRNAs had expression less than 10, and the only highly expressed circRNA was novel-circ-007090, which was expressed at all four time points (the expression from D3 to Y3 is 13,164, 9,931, 22,328, and 12,416, respectively), and its function needs further study.

GO analysis of ncRNA found that its main enriched GO terms for biological processes were cellular process and biological regulation, its main enriched GO terms for cellular components were cells, cell parts and organelle (except for circRNA), and its main enriched GO terms for molecular functions were binding and catalytic activity. In addition, KEGG analysis of circRNA showed that the pathway was mainly enriched in cellular community-eukaryotes, cell growth and death. These results are consistent with those of some previous studies, and in the study of skin development in mice, related terms, such as cell part, organelle, binding and catalytic activity, have also been found to be enriched (*Fore, 2006*;

*Zhu et al., 2020*). This suggests that these terms and pathways may indeed be associated with skin growth and development.

Studies have shown that lncRNAs and circRNAs can adsorb miRNAs in the form of sponges to regulate the expression of target genes (*Chen et al., 2019*; *Chen et al., 2018*; *Tay, Rinn & Pandolfi, 2014*). Therefore, we established miRNA-centered lncRNA–miRNA, miRNA–mRNA, and circRNA–miRNA targeting relationship pairs and constructed corresponding networks based on these relationship pairs. In the lncRNA–miRNA–mRNA coexpression network, ssc-miR-615 is associated with cell proliferation and apoptosis (*Tang et al., 2022*; *Wu et al., 2020*). ssc-miR-9820-5p can be adsorbed by circSLC41A1 in the form of sponge adsorption to promote SRSF1 and thus resist apoptosis of porcine granule cells (*Wang et al., 2022*). Cyclin dependent kinase inhibitor 2B (CDKN2B) is associated with cell growth and death (*Pan et al., 2021*; *Yang et al., 2022*). Dual specificity phosphatase 7 (DUSP7) is associated with development and regeneration (*Guo et al., 2021*). In the circRNA-miRNA network, we predicted the target genes of the top 20 circRNAs with the highest expression and obtained the corresponding relationship networks, but since circRNAs are all unknown, their interactions need to be further explored.

## CONCLUSIONS

In this study, we studied mRNAs, lncRNAs, miRNAs, and circRNAs in the skin at different developmental stages, and screened some genes associated with skin development. GO and KEGG analyses were used to perform functional analysis of differentially expressed genes and construct corresponding ceRNA networks. This study provides a reference for further research on pig skin development.

## ACKNOWLEDGEMENTS

The authors would like to thank Chengdu Livestock and Poultry Genetic Resources Protection Center for providing experimental animals. RNA sequencing was performed at BGI Genomics, BGI-SHENZHEN, China.

### Funding

This study was supported by the Key R&D Program of Sichuan Province (2020YFN0018) and the Chengdu Livestock and Poultry Genetic Resources Protection Center (2022). The funders had no role in study design, data collection and analysis, decision to publish, or preparation of the manuscript.

### Competing Interests

The authors declare there are no competing interests.

### Author Contributions

- Yujing Li conceived and designed the experiments, performed the experiments, analyzed the data, prepared figures and/or tables, authored or reviewed drafts of the article, and approved the final draft.

- Rui Shi conceived and designed the experiments, analyzed the data, prepared figures and/or tables, authored or reviewed drafts of the article, and approved the final draft.
- Rong Yuan performed the experiments, authored or reviewed drafts of the article, and approved the final draft.
- Yanzhi Jiang conceived and designed the experiments, analyzed the data, authored or reviewed drafts of the article, and approved the final draft.

## Ethics

The following information was supplied relating to ethical approvals (i.e., approving body and any reference numbers):

The University of Sichuan Agricultural granted Ethical approval to carry out the study within its facilities (Ethical Application Ref: 20220279).

## Data Availability

The data is available at NCBI GEO: GSE231573.

## Supplemental Information

Supplemental information for this article can be found online at http://dx.doi.org/10.7717/peerj.15955#supplemental-information.

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
