# Peer review of "Comprehensive transcriptional analysis of pig facial skin development"

_PeerJ, doi:10.7717/peerj.15955_

## Round 0.1 · original submission · Minor Revisions

Two reviewers (Reviewer 1 and 3) identified some issues that require further comments, including things as straightforward as the difference between facial skin and the more typically compared skin of the back. Please address the comments by these two reviewers thoroughly.

·

Basic reporting

Literature references, sufficient field background/context provided

Experimental design

Research question well defined, relevant & meaningful. It is stated how research fills an identified knowledge gap.

Validity of the findings

All underlying data have been provided; they are robust, statistically sound, & controlled

Additional comments

In this study, the facial skin samples of Chenghua Chinese pigs at four different developmental stages were used as experimental materials. Through pairwise comparison of the four developmental stages, it was found that the number of differentially expressed RNAs(DEGs) increased with increasing developmental time intervals, and elastin (ELN) was found to be a key factor. PPAR signaling pathway, which is the classic pathway of skin development, was enriched. In addition, the interaction networks of lncRNA-miRNA-mRNA and circRNA-miRNA were constructed, which laid a foundation for understanding the specific mechanism of pig skin development. Overall, this study is well presented, but there are some minor concerns which need to be addressed below:

Minor concerns
-There are many papers on pig skin as xenotransplantation materials or biological dressings, and most of them are taken from the back of pigs. In this experiment, the skin of pig face is taken. Please describe in the discussion whether there is any significant advantage of the skin of this site over the back.
-Processes that require supplemental predicted lncRNAs targeting mRNAs (prediction methods).
- When performing miRNA functional enrichment analysis, should the authors conduct GO and KEGG analysis after relevant prediction? If so, please describe the prediction process and clarify the parameters in the previous section.
- The author only verified the expression of some genes in RNA-seq in the final verification stage, and did not verify the obtained ceRNA network. Should relevant cell experiments be carried out to verify it?

Reviewer 2 ·

Basic reporting

This manuscript meets the standards of the journal.

Experimental design

no comment

Validity of the findings

no comment

Additional comments

I believe the results of the study are valid and should be published without significant change

·

Basic reporting

The scientific paper appears well organized with authors providing a clear introduction, background and description of the research question and the rationale for the study.

However, there are few areas the authors could improve the clarity of reporting.

Few comments and recommendations:
1) In some places, the language is quite technical and could be made more accessible to readers who are not experts in the field. For eg. Acronyms should be mentioned in full wherever possible & the first time they are used in the paper (e.g PPAR. DNB, UMI, etc)
2) In Table S1, please specify what the different RNAs are especially mRNAs and lncRNAs - they are specified later in the paper, but good to have with the table as well.
3) Figure 1B and 1C is very difficult to interpret. Even zooming in did not help. Update the figures with better resolution.
4) Line 246, it mentions Venn diagram - the figure 3A is infact a Upset diagram.
5) Line 264 - Figure S4 provided consists of D180/D90 and Y3/D180 data comparison. The data for D3 vs D180 seems missing.
6) Figure 3B and 4B is similar
7) Line 311 - Typo mistake

Experimental design

The experimental design is very well described, including the methods used to collect and analyze the data. The information is very thorough backed up with supplemental data which makes it reproducible by any other investigator.

Validity of the findings

The study provides valuable insights into the development of skin in postnatal Chenghua pigs and the regulatory mechanisms involved which provides a stepping stone for further research on the pig skin development. The authors have presented their findings clearly, and the results are well supported by data.

However the authors can provide more details on the following:
1) Considering animal to animal variability - it is important to justify the selection of 3 animals/group & confidence in the scientific findings - this can help the author discuss the limitations of their study.
2) Provide more information on the significance of the identified DEGs and ncRNAs and their potential use in skin research.
3) Suggest future directions for research in this area.

Additional comments

Overall, the paper is well-written and effectively communicates the author's findings.

---

## Round 0.2 · accepted · Accept

Thank you for your efforts to address the comments of the reviewers. Excellent work!